# Elevated Interleukin-31 Levels in Serum, but Not CSF of Dogs with Steroid-Responsive Meningitis-Arteritis Suggest an Involvement in Its Pathogenesis

**DOI:** 10.3390/ani13162676

**Published:** 2023-08-20

**Authors:** Laura Lemke, Regina Carlson, Katrin Warzecha, Andrea V. Volk, Andrea Tipold, Jasmin Nessler

**Affiliations:** Department of Small Animal Medicine and Surgery, University of Veterinary Medicine Hannover, 30559 Hannover, Germany; laura.lemke@tiho-hannover.de (L.L.); katrin.warzecha@tiho-hannover.de (K.W.); andrea.volk@tiho-hannover.de (A.V.V.); andrea.tipold@tiho-hannover.de (A.T.)

**Keywords:** IL-31, SRMA, canine, Th-2 mediated inflammation, MUO, meningoencephalitis of unknown origin

## Abstract

**Simple Summary:**

Steroid-responsive meningitis-arteritis (SRMA) is a common immune-mediated inflammatory disease of young-adult dogs with predominantly neck pain and fever. The exact pathomechanism of this disease remains unclear. Interleukin-31 is a cytokine that has been shown to be elevated among other things in dogs with atopic dermatitis and dogs with a typical phenotype of inflammation (Th2-mediated) in previous studies. This phenotype of inflammation is typical for SRMA. Therefore, we suggested that interleukin-31 might be involved in the pathomechanism of SRMA. Within the study, interleukin-31 levels were measured in serum and cerebrospinal fluid samples of dogs affected by SRMA in comparison to dogs with atopic dermatitis (positive control), healthy dogs (negative control) and dogs with immune-mediated or infectious meningoencephalitis. The serum interleukin-31 levels of dogs with SRMA were markedly higher than in healthy control dogs. Especially dogs with SRMA without any pre-treatment showed markedly higher interleukin-31 levels. The cerebrospinal fluid interleukin-31 levels revealed no differences between the groups. Based on this study, an involvement of interleukin-31 in the pathogenesis of SRMA can be assumed, but further clarification is necessary with prospective studies. We suggest that our study might help to clarify further parts of the pathogenesis of SRMA.

**Abstract:**

Steroid-responsive meningitis-arteritis (SRMA) is a predominantly Th-2 immune-mediated disease, but the exact pathomechanism remains unclear. Interleukin-31 (IL-31) is predominantly produced by T cells with a Th-2 phenotype during proinflammatory conditions. We hypothesize that IL-31 might be involved in the pathogenesis of SRMA. IL-31 was measured in archived samples (49 serum and 52 CSF samples) of dogs with SRMA, meningoencephalitis of unknown origin (MUO), infectious meningoencephalitis, and atopic dermatitis, and of healthy control dogs using a competitive canine IL-31 ELISA. The mean serum IL-31 level in dogs with SRMA (*n* = 18) was mildly higher compared to dogs with atopic dermatitis (*n* = 3, *p* = 0.8135) and MUO (*n* = 15, *p* = 0.7618) and markedly higher than in healthy controls (*n* = 10, *p* = 0.1327) and dogs with infectious meningoencephalitis (*n* = 3, no statistics). Dogs with SRMA in the acute stage of the disease and without any pre-treatment had the highest IL-31 levels. The mean CSF IL-31 value for dogs with SRMA (*n* = 23) was quite similar to that for healthy controls (*n* = 8, *p* = 0.4454) and did not differ markedly from dogs with MUO (*n* = 19, *p* = 0.8724) and infectious meningoencephalitis. Based on this study, an involvement of IL-31 in the pathogenesis of the systemic Th-2 immune-mediated immune response in SRMA can be assumed as a further component leading to an aberrant immune reaction.

## 1. Introduction

Steroid-responsive meningitis-arteritis (SRMA) describes a systemic immune-mediated inflammatory reaction affecting mostly the cervical spinal meninges, leading to a suppurative leptomeningitis with severe arteritis [1]. It typically occurs in young-adult dogs, and clinical signs include stiff gait or reluctance to walk, pyrexia, and neck pain worsening during neck flexion or extension in acute cases [1,2,3,4]. In some cases, SRMA occurs in combination with an immune-mediated polyarthritis [2,4,5]. Typical findings in blood work of dogs with SRMA are leukocytosis with neutrophilia and left-shift; in cerebrospinal fluid (CSF) samples, a neutrophilic pleocytosis is visible [1,3,4]. Additionally, in dogs with SRMA, elevated immunoglobulin A (IgA) levels in serum and CSF are frequently measurable [6]. Immunosuppressive or anti-inflammatory therapy with prednisolone or with prednisolone in combination with other immunosuppressive drugs often has a positive outcome [1,3,4,7,8,9], but relapses can occur in up to one-half of treated dogs with SRMA [3,4,7]. A T-helper cell 2 (Th2) mediated immune response is highly probable in SRMA [10,11].

In one of our previous studies, it could be shown that interleukin-31 (IL-31) levels were elevated in serum samples of dogs with suspected Th 2-helper cell-mediated inflammatory response, including dogs with secondary inflammation after an intervertebral disc herniation or with otitis media and interna [12]. Interleukin-31 is a four-helix bundle cytokine from the gp130/IL-6 cytokine family [13,14]. It is mainly produced by activated Th2 cells, especially from CD4+ T cells [13,14,15]. IL-31 develops its effect by binding to a receptor complex consisting of the IL-31 receptor A and the oncostatin M receptor β [14]. IL-31 has an important role in inducing atopic dermatitis and pruritus in several species [13,16,17,18,19]. In addition, its influence on various immune and non-immune cells and regulation of immune responses and hematopoiesis was studied [14]. IL-31 is involved in developing inflammatory bowel disease, airway hypersensitivity and dermatitis [14].

As described above, IL-31 is part of the interleukin-6 (IL-6) cytokine family and expressed during proinflammatory conditions. According to Maiolini et al., interleukin-6 (IL-6) is elevated in patients with SRMA and was correlated with the degree of CSF pleocytosis [20]. We hypothesize that IL-31 might be involved in the pathogenesis of SRMA represented by elevated IL-31 levels in the serum and/or CSF of dogs with SRMA. Therefore, we hypothesize that IL-31 levels in the serum and/or CSF of dogs with SRMA are comparable to those in samples from dogs with atopic dermatitis and higher compared to the levels in samples from dogs with other types of inflammatory CNS diseases (e.g., MUO or infectious meningoencephalitis). In addition, we expect a decline in IL-31 levels in the serum and/or CSF of dogs with SRMA after treatment with glucocorticosteroids. The aim of this study is twofold: firstly, to clarify a further part of the pathogenesis of SRMA and the effect of pre-treatment on the IL-31 levels, and secondly, to enable a new therapeutic approach for a specific treatment against IL-31, provided that our hypothesis is confirmed.

## 2. Materials and Methods

This study was conducted as a retrospective single center study. The clinic’s biobank was searched for archived serum and CSF samples from dogs with SRMA, meningoencephalitis of unknown origin (MUO), infectious meningoencephalitis, or atopy presented between 2011 and 2021 at the Department of Small Animal Medicine and Surgery, University of Veterinary Medicine Hannover (Hannover, Germany). All patient samples were obtained as part of routine diagnostic examination and used with the owners’ written consent. The samples from healthy beagles were residuals of previous studies performed according to the ethical guidelines of the University of Veterinary Medicine Hannover and approved by the Lower Saxony State Office for Consumer Protection and Food Safety (Lower Saxony, Germany; animal experiment number 33.12-42502-04-20/3352). All samples were stored after sampling at −80 °C.

General information about breed, age, sex, onset and duration of clinical signs, previous treatment and results of examinations and diagnostic tests, including general and neurological examination, blood examination, magnetic resonance imaging (MRI) of the brain and/or cervical/thoracic spinal cord if available, CSF examinations, and IgA content in CSF and serum, were taken from the electronic medical record of each individual patient (easyVET, Veterinärmedizinisches Dienstleistungszentrum (VetZ) GmbH, Isernhagen, Germany).

The samples from the dogs were grouped according to their underlying diseases into five groups:Group A: Patients with SRMA (*n* = 26). The diagnosis of SRMA was based on typical clinical signs of SRMA, including neck pain detectable by stiff gait, low head carriage, vocalization occurring spontaneously or during the neurological examination, anamnestic pyrexia or fever during the clinical examination in combination with blood leukocytosis and a moderate to marked predominantly neutrophilic pleocytosis in the cerebrospinal fluid, as described in the current literature [1,2,3,4].Group B: Patients with MUO (*n* = 21). In 14 dogs, the presumptive diagnosis of MUO was made on the basis of clinical signs of intracranial lesions in combination with MRI of the brain showing intra-axial, T2-weighted hyperintense and T1-weighted iso- to hypointense lesions with a variable degree of contrast enhancement in T1-weighted sequences [21,22,23] in combination with the result of a CSF examination showing a lymphocytic or mononuclear or mixed pleocytosis with an elevated protein content [22]. Infectious meningoencephalitis was ruled out by negative testing for frequently occurring pathogens in serum and CSF (*n* = 14). In seven patients, the diagnosis of MUO was histopathologically confirmed, and all cases were sub-classified as granulomatous meningoencephalitis (*n* = 7).Group C: Patients with infectious meningoencephalitis (*n* = 3). Two cases revealed bacterial meningoencephalitis (*n* = 2), and one dog a lymphoplasma-histiocytic meningoencephalomyelitis with suspected viral etiology. The diagnosis was histopathologically confirmed in two cases.Group D: Patients with atopic dermatitis (*n* = 3) served as positive control group. Diagnosis of atopic dermatitis was suspected after exclusion of other possible underlying causes for itching like cutaneous infection or ectoparasites, if the dogs showed initial itching without lesions at the age of less than 3 years, had an indoor lifestyle, and affected areas were feet and concave aspects of the pinnae [24]. For these patients, only serum samples were available and included in this study.Group E: Healthy control group (*n* = 11). This group included samples from healthy clinic-owned beagles. The general clinical examination was unremarkable in these dogs, and all included dogs showed no clinical signs of atopic dermatitis.

Group A patients with SRMA were further divided into four subgroups considering pre-treatment before the sampling was performed. The pre-treatment groups were divided into no prior treatment (group A.1), pre-treatment with non-steroidal anti-inflammatory drugs (NSAIDs) (group A.2), pre-treatment with metamizole (group A.3), and pre-treatment with steroids (group A.4). IL-31 levels in the serum of dogs with SRMA were further subdivided into the following parts: extremely high IL-31 levels above the maximum IL-31 level of dogs with atopic dermatitis serving as a positive control group, low IL-31 levels below the mean IL-31 level of the healthy control animals serving as a negative control group; the remaining samples had values between both borders.

The obtained CSF and serum samples were examined for the IL-31 level using a competitive enzyme-linked immunosorbent assay (ELISA; Canine Interleukin 31 ELISA kit, MyBioSource, Inc., San Diego, CA, USA, catalogue number: MBS740462). The frozen samples were thawed immediately before the assay was performed and brought to room temperature according to the manufacturer’s instructions. Subsequently, the ELISA was carried out according to the enclosed instructions of the manufacturer. The measurement of the optical density of the samples was carried out immediately after the addition of the stop solution from the ELISA kit in a multi-detection microplate reader (SynergyTM 2, BioTek Instruments, Inc., Winooski, VT, USA) at 450 nm. Each sample was measured in duplicates, and the coefficient of variation was determined.

Only results from samples with a coefficient of variation below 20% were considered for the statistical analysis. The coefficient of variation was used as a marker for the intraassay variation for testing in duplicates of each sample [25]. If the coefficient of variation values were higher than 20% for one sample, the measurement was repeated. This was not possible for all samples due to a small sample size. If the coefficient of variation value was again higher than 20% in the second attempt or if the sample amount was too small for repeated measurements, these samples were excluded from statistical analysis. A standard reference curve was established using six standard solutions with a specified canine IL-31 level of 0–1000 pg/mL. If IL-31 values exceeded the upper detection range of 1050 pg/mL, values were given as >1050 pg/mL, and statistical analysis was performed with the fixed value of 1050 pg/mL.

The statistical evaluation was carried out using the GraphPad Prism 9 software (GraphPad Software (part of Dotmatics), San Diego, CA, USA). After evaluating descriptive statistics, the examination for normal distribution was carried out using the Shapiro–Wilk test. The results for CSF IL-31 levels were logarithmized to achieve a normal distribution to carry out further statistical tests. Afterwards, the Mann–Whitney or Kruskal–Wallis test and the unpaired t-test or one-way analysis of variance were used for testing for significance with a significance level of *p* < 0.05. In paired serum and CSF samples or to detect a correlation between two other parameters, the statistical evaluation of correlation was performed with the Spearman correlation.

## 3. Results

A total of 49 serum and 52 CSF samples from 64 patients with a coefficient of variation below 20% were used for the statistical evaluation. The sample distribution was as follows:26 patients with SRMA (group A)21 patients with MUO (group B)3 patients with infectious meningoencephalitis (group C)3 patients with atopic dermatitis (group D)11 patients within the healthy control group (group E)

As described above, IL-31 levels could not be evaluated in all samples due to an inappropriate coefficient of variation. Therefore, the numbers of samples included in statistical analysis differed from totally evaluated numbers. Beagle was the most frequently represented breed (*n* = 13/64; including 11 healthy Beagles of group E), followed by Boxer (*n* = 9/64), mixed breed dogs (*n* = 8/64), French Bulldogs (*n* = 4/64), Bernese Mountain Dogs and Yorkshire Terriers (each *n* = 3/64). Two samples from the following breeds each were included: Golden Retriever, Magyar Vizsla, Labrador Retriever, Pug Dog and Belgian Shepherd Dog. Additionally, one sample from the following breeds was included: Weimaraner, German shorthaired pointer, Nova Scotia Duck Tolling Retriever, German Pinscher, Goldendoodle, Border Collie, Chihuahua, German Shepherd, Podenco Ibicenco, Airedale Terrier, Australian Kelpie, Galgo Español, Husky and Hanoverian Scenthound. In the group with SRMA, Boxers were the most frequently occurring breed (*n* = 9), followed by Bernese Mountain Dogs (*n* = 3). Further, 45.3% of the dogs were male (*n* = 29/64), and 14.1% of the dogs were male-neutered (*n* = 9/64). In addition 31.2% of the dogs were female (*n* = 20/64), and 9.4% female-spayed (*n* = 6/64). The median age of all dogs was 43.2 months (*n* = 64), with a range from 3 to 120 months.

### 3.1. Serum Samples

In the serum samples, healthy control dogs had IL-31 levels ≤ 137.2 pg/mL. The lowest mean IL-31 level was measured in serum samples from the healthy control group. The IL-31 levels in serum samples from dogs with infectious meningoencephalitis were comparable low to the levels in samples from the healthy control group (Table 1), but due to the low number of serum samples from dogs with infectious meningoencephalitis, no statistical analysis was performed. The highest mean IL-31 content was measured in samples from patients with SRMA (*n* = 18, Table 1), followed by the second highest mean IL-31 level in serum samples from the dogs with atopic dermatitis (*n* = 3, Table 1) and patients with MUO (*n* = 15, Table 1). Although IL-31 levels in serum from dogs with SRMA were markedly higher than the mean IL-31 level for the healthy control group, no statistically significant difference could be demonstrated (*p* = 0.1327). The mean IL-31 level in patients with SRMA was mildly higher than in patients with atopic dermatitis (*p* = 0.8135) and MUO (*p* = 0.7618), but this difference was not statistically significant. IL-31 levels in serum from dogs with MUO was significantly higher than in serum from healthy dogs (*p* = 0.0163). Complete descriptive statistics are presented in Table 1, and IL-31 levels in serum samples from the different groups A to E are visualized in Figure 1A.

IL-31 levels varied widely within the serum samples from dogs with SRMA. Four patients with SRMA had extremely high IL-31 levels above 437.9 pg/mL (corresponding to the maximum IL-31 level in dogs with atopic dermatitis serving as a positive control group), and six dogs with SRMA revealed relatively low IL-31 levels in serum below 80.73 pg/mL (corresponding to the mean IL-31 level in the healthy control serving as a negative control group). The remaining eight results were in between the range between >80.73 pg/mL and <437.9 pg/mL. An IL-31 level > 1050 pg/mL was measured in the serum from one patient in group A with SRMA. The exact value of this sample could not be determined, as a small sample size prevented repeated measurement of the diluted sample. Almost all dogs from the group with high IL-31 values > 437.9 pg/mL in serum did not receive any pre-treatment before sample collection (*n* = 3/4). Only one dog received pre-treatment with metamizole prior to sampling (*n* = 1/4). Patients with very high (*n* = 4) or moderately elevated serum IL-31 levels (*n* = 8) were predominantly patients with a peracute or acute disease process, and duration of clinical signs was 3 days or less in these dogs (*n* = 8/12). In addition, SRMA patients with very high IL-31 levels in serum showed increased body temperature (*n* = 3/4; temperature unknown *n* = 1/4). In contrast, SRMA patients with low serum IL-31 levels had predominantly shown clinical signs of the disease for 5 days or longer (*n* = 5/6). In addition, the majority of dogs with SRMA and low IL-31 levels in serum (*n* = 6) showed a physiological body temperature or a mildly, subfebrile elevation of the body temperature below 39.3 °C at the physical examination prior to sampling.

### 3.2. CSF Samples

In the CSF, the mean IL-31 level of patients with SRMA (*n* = 23) and the healthy control group (*n* = 8) did not differ significantly (*p* = 0.4454), whereas the mean IL-31 content in CSF samples from patients with MUO (*n* = 19) was mildly higher compared to patients with SRMA (*p* = 0.8724) and to the healthy control group (*p* = 0.5253, Table 1). The mean IL-31 level in CSF samples from patients with infectious meningoencephalitis (*n* = 3) was higher than in the other groups, but due to the low number of CSF samples from dogs with infectious meningoencephalitis, no statistical analysis was performed. No CSF samples from dogs with atopic dermatitis were included in this calculation. All results for the IL-31 content in CSF samples from dogs in the different groups A to E are visualized in Figure 1B.

### 3.3. Testing for Correlation

For 35 dogs in total and for 15 dogs with SRMA, IL-31 levels in CSF and serum were available from the same sampling time point. IL-31 levels in CSF and serum were not correlated when taking dogs of all groups into consideration (*n* = 35; r = 0.05191, *p* = 0.7349), as well when just looking at dogs with SRMA (*n* = 15; r = −0.05714, *p* = 0.8425). No correlation was detectable between the IL-31 levels in the serum or CSF samples from dogs with SRMA and the amount of leukocytes in CSF (r = 0.1502, *p* = 0.4939). In addition, there was no correlation between the protein content in the CSF samples with the IL-31 level in serum (r = −0.03612, *p* = 0.8869) or CSF of dogs with SRMA (r = 0.06991, *p* = 0.7513). IgA levels were available in the serum of 17 dogs with SRMA (*n* = 17/18) and in CSF of 18 dogs with SRMA (*n* = 18/23), but there was also no correlation between the IgA levels in serum and CSF and the IL-31 levels in serum (r = 0.07108, *p* = 0.7874) and CSF (r = 0.1889, *p* = 4529).

### 3.4. Pre-Treatment

To evaluate whether IL-31 levels in dogs with SRMA (group A) might be influenced by a prior treatment, this group was divided into four subgroups depending on the pre-treatment. Seven dogs with SRMA received no prior treatment (group A.1, *n* = 7). Seven dogs were pre-treated with NSAIDs (group A.2, *n* = 7) including treatment with carprofen (*n* = 3) or meloxicam (*n* = 4). No information regarding the dosage was available. Six dogs were pre-treated with metamizole (group A.3, *n* = 6). Four dogs received steroids as a prior treatment (group A.4, *n* = 4). Two dogs received prednisolone, one dog an unknown glucocorticosteroid, and one dog dexamethasone and prednisolone together. For two dogs, the pre-treatment was unknown, so they were excluded from the statistical analysis.

The mean IL-31 content in serum was highest in dogs with SRMA without prior treatment (group A.1, *n* = 5), followed by dogs with SRMA pre-treated with metamizole (group A.3, *n* = 4), but the mean IL-31 level was less than half of the mean IL-31 level in the group without treatment (Figure 2). In the serum of dogs with SRMA pre-treated with NSAIDs (group A. 2, *n* = 4) and with steroids (group A.4, *n* = 3), the mean IL-31 content was similar, but markedly lower compared to both of the other groups (Figure 2). Statistically, no significant differences were detectable within the serum samples (*p* > 0.05).

Regarding CSF samples, the results were more homogenous between the groups and different from the serum results. The highest IL-31 levels were measured in the CSF of dogs with SRMA pre-treated with steroids (group A.4, *n* = 4), followed by the mean IL-31 level in dogs with SRMA pre-treated with NSAIDs (group A.2, *n* = 6) (Figure 2). The lowest IL-31 levels were detected in CSF samples from dogs pre-treated with metamizole (group A.3, *n* = 5) and dogs with SRMA without pre-treatment (group A.1, *n* = 7, Figure 2). The IL-31 levels were significantly higher in the CSF of dogs with SRMA pre-treated with steroids (group A.4, *p* = 0.0350) and of dogs with SRMA pre-treated with NSAIDs (group A.2, *p* = 0.0424) compared to dogs without pre-treatment. No further significant differences were detectable within the CSF samples (*p* > 0.05).

## 4. Discussion

The purpose of this study was to investigate the presence of IL-31 in serum and CSF of patients with SRMA and their dependence on a possible pre-treatment. We expected that dogs with SRMA would have higher levels of IL-31 in their CSF and/or serum due to the Th2-mediated immune response in SRMA [10,11] and hypothesized that IL-31 might be involved in the pathogenesis of SRMA as a further component leading to an aberrant immune reaction in SRMA. In SRMA, several acute-phase proteins are elevated, indicating an inflammatory reaction [26]. Especially, elevation of the C-reactive protein (CRP) can be measured in the in serum and CSF of acute SRMA cases [26,27], and a significant decrease was found in serum and CSF after treatment with prednisolone [26]. In addition, the immunoglobulin A (IgA) levels were elevated in serum and CSF in SRMA [6,28], but did not decrease after starting an immunosuppressive therapy or during remission [3,6,7].

IL-31 is a cytokine produced by activated Th2 cells and influences the immune response [13,14]. In one of our previous but unpublished studies, it could be shown that IL-31 levels were elevated in serum samples from dogs with a suspected Th2 cell-mediated inflammatory response, including dogs with secondary inflammation after intervertebral disc herniation or with suspected otitis media and interna ([12] unpublished data).

Indeed, it could be shown that patients with SRMA had increased IL-31 levels in serum. Especially when comparing the IL-31 levels of dogs with SRMA (group A) with the healthy control group (group E), the IL-31 levels were markedly higher in the serum of SRMA patients. In CSF samples, the mean IL-31 levels were quite similar between the groups, so our hypothesis that higher IL-31 levels would be detected in CSF samples from SRMA cases could not be confirmed with the measured data of this study, but there was an elevation of IL-31 levels in the CSF after steroidal treatment, which will be discussed later.

Particularly high levels of IL-31 were observed in serum of SRMA patients with a peracute or acute disease process of 3 days or less, whereas SRMA patients with low serum IL-31 levels had predominantly shown clinical signs of disease for at least 5 days. However, it must be taken into account that the duration of the clinical signs was calculated based on the information provided by the owners, and that an objectively measured time period was not available. In addition, different medications had been administered to some of the animals in the period between the onset of clinical symptoms and the sampling. Therefore, an exact correlation between disease onset and IL-31 levels could not be calculated.

In addition, patients with very high IL-31 levels in serum showed increased body temperature in the clinical examination. Fever can be caused by infectious diseases via exogenous pyrogens or neoplasms, but in nearly half of dogs, fever is caused by non-infectious inflammatory diseases [29,30]. Interleukin-1 (IL-1) is an endogenous pyrogen, which—in addition to inducing fever—also leads to activation of B and T cells to be able to generate an effective immune response to the causative pathogen [29]. IL-1 also stimulates the production of IL-6 [31,32], which is a pro-inflammatory cytokine activating mechanisms for fever in the central nervous system (CNS) [29,32]. In addition, IL-6 also induces the production of acute-phase proteins in the liver like CRP [29]. As already described, IL-31 is part of the IL-6 cytokine family [13,14], which could well explain the high IL-31 serum levels in dogs with SRMA and a higher body temperature. But in contrast to IL-6 [20], IL-31 seems not be involved in the migration of cells into the CSF space, as IL-31 levels did not correlate with the degree of CSF pleocytosis in the current study.

The breed distribution of the group of dogs with SRMA was in line with previous studies describing a higher prevalence in Boxers and Bernese Mountain dogs amongst others [1,4]. The typical findings of SRMA in CSF, including a neutrophilic pleocytosis, were also detected in most of the dogs with SRMA in this study.

IL-31 levels in CSF varied in dogs with SRMA. Due to the retrospective nature of this study, the site of CSF sampling was documented for all patients. It might be possible that in some dogs, lumbar, while in others, suboccipital tap was performed. The sampling site can have an impact on the results of CSF examinations, leading to false negative results in up to 7% of dogs with SRMA [33], and might also have an impact on IL-31 levels.

We also included dogs with MUO in this study for comparison with another inflammatory disease of the CNS. MUO is also an immune-mediated inflammatory condition, but in this disease, not only the meninges, but also the brain and/or the spinal cord are affected [21,22,23]. In addition, the group of patients with infectious meningoencephalitis was included in this study in order to compare with a non-immune-mediated inflammatory CNS disease. However, due to the insufficient number of patients with infectious meningoencephalitis, no further statistical analysis was performed regarding this group.

While MUO is an umbrella term, it includes several histologic subtypes like granulomatous meningoencephalitis (GME) or necrotizing meningoencephalitis [22]. In the current study, not all patients with MUO had histopathological examinations. The ones that did were classified as GME. Patients with MUO had significantly higher IL-31 levels in serum compared to the healthy control group, but lower IL-31 levels in serum compared to dogs with SRMA. Although MUO is a predominantly Th1-driven immune reaction, the expressed patterns of cytokines and chemokines differ between the subtypes of MUO and even between individual patients [34], and especially in GME, increased Th17 cells indicate a contribution of an additional Th2-immune response similar to that of SRMA [34,35,36]. Therefore, our results are in line with previous findings of a predominantly Th2-driven immune response in SRMA. Patients with MUO also had elevated IL-31 levels compared to the healthy control group in this study, which could support a mixed Th1/Th2-immune response in MUO patients. Further investigation of the elevated IL-31 levels in serum and mildly higher IL-31 levels in CSF of patients with MUO compared to dogs with SRMA is required, especially in regard to a more detailed investigation of the different subtypes of MUO. This requires larger study populations with histopathologically confirmed diagnoses of MUO subtypes. However, this goes beyond the aims and investigations carried out in this study, including MUO as a control population.

Three dogs with infectious meningoencephalitis were also included in this study. The IL-31 level in serum was low compared to that in the healthy control group, but in the CSF a higher IL-31 level was detected. In contrast to SRMA, which is a systemic immune-mediated inflammatory disease [1], the dogs had a more local infectious inflammatory reaction in the CNS caused by bacterial infection either secondary to ascending otitis or iatrogenic bacterial infection in two out of the three dogs. Due to the blood–brain barrier, which prevents the uncontrolled transfer of cells from the blood into the CNS parenchyma [37], activated immune cells and their cytokines might not be visible in the peripheral blood in cases of infectious meningoencephalitis. This could explain the low IL-31 levels in the serum of dogs with infectious meningoencephalitis and the elevated IL-31 levels within the CSF of these patients.

In the current study, we also examined the influence of different pre-treatments on the IL-31 levels at the time point of diagnosis of SRMA. The results revealed markedly higher levels of IL-31 in serum of dogs without any prior treatment. Dogs with SRMA pre-treated with NSAIDs or prednisolone showed markedly lower IL-31 levels in serum than dogs without any prior treatment, but the serum IL-31 levels were still above the mean levels in serum of the healthy control group. Immunosuppressive/anti-inflammatory therapy with prednisolone monotherapy is mostly successful in dogs with SRMA [1,3,7], but sometimes a combination with further immunosuppressive medication is necessary [8,9]. Among other mechanisms, prednisolone reduces the release of IL-1 and IL-6, which is one reason for its anti-inflammatory effect [38]. In addition, a direct suppression of the function of T cells can be assumed [38]. Dogs pre-treated with prednisolone, therefore, might have lower IL-31 levels in the serum due to the effects of prednisolone. NSAIDs are anti-inflammatory and analgesic drugs [39]. They are not useful to treat SRMA in a long-term scheme, but as a pre-treatment, they can lead to a suppression of the inflammatory reaction and might, therefore, lower IL-31 levels. In contrast, metamizole is an analgesic and anti-pyretic drug, with only limited anti-inflammatory effect [39]. Therefore, it does not suppress IL-31, which might explain the higher IL-31 levels in serum of dogs with SRMA pre-treated with metamizole in comparison to dogs pre-treated with prednisolone or NSAIDs. However, due to the retrospective nature of this study, the administration of prednisolone was quite variable prior to the initial sampling of serum and CSF to diagnose SRMA.

Results regarding IL-31 levels in CSF of pre-treated dogs with SRMA were unexpected. In CSF, the highest IL-31 levels were detected in dogs with SRMA pre-treated with prednisolone, followed by dogs with SRMA pre-treated with NSAIDs, while the lowest IL-31 levels were measured in dogs with SRMA without any pre-treatment. Due to the severe arteritis occurring with SRMA and due to an upregulation of metalloproteinases (MMPs), including MMP-2 and -9, a disruption of the blood–brain barrier occurs [40]. In combination with an upregulation of integrins like CD11a in the CSF, an invasion of neutrophils into the subarachnoid space leads to marked neutrophilic pleocytosis in the CSF [41]. During this invasion into the subarachnoid space, other blood cells also can move into the subarachnoid space and into the CSF, such as Th2 cells producing IL-31. Treatment with glucocorticosteroids restores the integrity of the blood–brain barrier [42], and additionally, glucocorticoid treatment leads to production of an MMP inhibitor [43], as MMPs have been shown to be one of the causes leading to blood–brain barrier disruption [40]. After treatment with glucocorticosteroids, the integrity of the blood–brain barrier, therefore, is increased, so that the immune cells might be trapped within the CSF still producing their interleukins. It is likely that the elevated IL-31 levels in the CSF of dogs with SRMA pre-treated with prednisolone represent the intrathecally produced IL-31 from Th2 cells remaining in the CSF after restoration of the blood–brain barrier. The same effect, but to a smaller degree, can be detected in dogs pre-treated with NSAIDs, due to the anti-inflammatory effect of this medication [39]. However, the survival time of immune cells in the CSF is short, since it is physiologically a very low-protein fluid [44]. A first decrease in immune cells in the CSF can be observed in vitro already after 1 h, but the granulocytes are affected first [44]. Therefore, it must also be considered that there was no further intrathecal IL-31 production at the time of sampling and that only the trapped IL-31 in CSF could be measured. This would require further studies regarding the number of IL-31 producing Th2 cells in the CSF in connection with the respective IL-31 level in the CSF during the course after treatment.

As already mentioned above, this study had a few limitations due to its retrospective nature. No standardized protocol was used regarding time points of serum and CSF sampling or regarding drug administration. No homogenous treatment group could formed, since some dogs received more than one drug simultaneously. Additionally, the diagnosis of MUO was confirmed histopathologically in only one-third of the included dogs, and the exact subtype of MUO (e.g., granulomatous meningoencephalitis) was not available for all cases. A further limitation of this study was the small sample size after dividing the dogs according to their pre-treatment.

## 5. Conclusions

Based on this study, an involvement of IL-31 in the pathogenesis of the Th-2 mediated immune response in SRMA and also in MUO can be assumed; in particular, the systemic changes in SRMA like generalized arteritis and the proinflammatory status (pyrexia, leukocytosis) can be well explained in SRMA. In the CSF, no elevated IL-31 levels in dogs with SRMA or MUO were detectable. Prospective, standardized studies examining IL-31 levels at different time points over the course of SRMA might be helpful to clarify IL-31 production during different stages of the disease and might help to design new therapeutic approaches in dogs with SRMA. Additionally, prospective studies examining the influence of treatment on IL-31 levels with standardized treatment protocols are necessary to clarify the progression of IL-31 levels under treatment.

## Figures and Tables

**Figure 1 animals-13-02676-f001:**
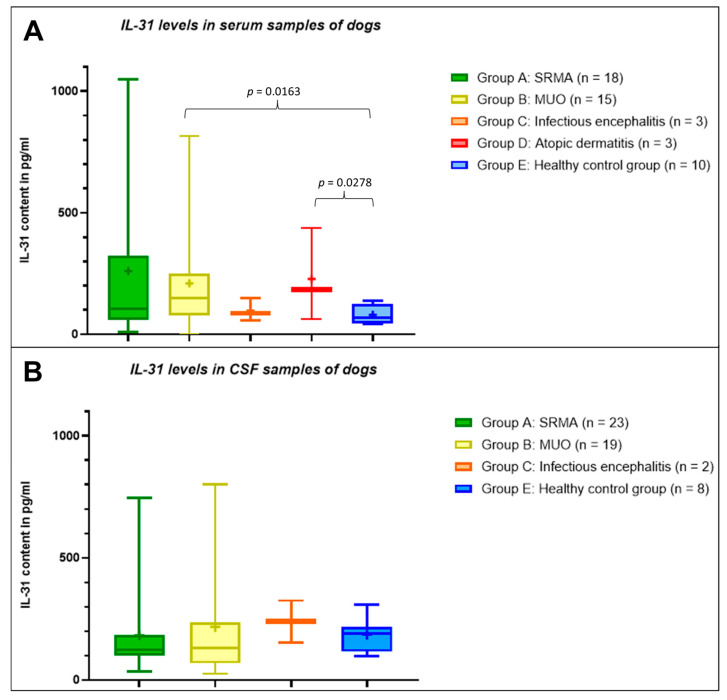
Interleukin-31 (IL-31) levels in serum (**A**) and cerebrospinal fluid (CSF) (**B**) samples from dogs. There is a significant difference within the serum samples between groups B and E (*p* = 0.0163) and groups D and E (*p* = 0.0278). The values for the other groups or for the CSF samples did not differ significantly (*p* > 0.05). The boxplots show whiskers from the minimum to maximum and boxes from the 25th to 75th percentiles as well as the median (horizontal line) and the mean (+). SRMA: steroid-responsive meningitis-arteritis; MUO: meningoencephalitis of unknown origin; pg/mL: picograms per milliliter.

**Figure 2 animals-13-02676-f002:**
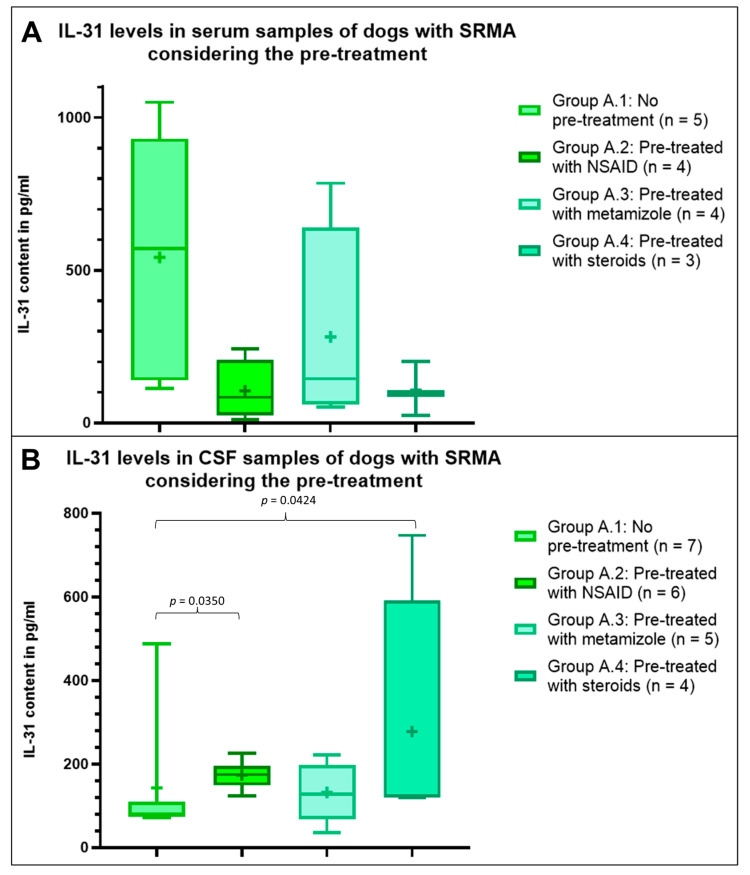
Interleukin-31 (IL-31) levels in serum (**A**) and cerebrospinal fluid (CSF) (**B**) samples from dogs with steroid-responsive meningitis-arteritis (SRMA) considering their pre-treatment. There was a significant difference within the CSF samples between patients pre-treated with non-steroidal anti-inflammatory drugs (NSAIDs; group A.2) and dogs without a prior treatment (group A.1; *p* = 0.0350) and between dogs pre-treated with steroids (group A.4) and dogs without a prior treatment (group A.1; *p* = 0.0424). The values for other groups did not differ significantly (*p* > 0.05). The boxplots show whiskers from the minimum to maximum and boxes from the 25th to 75th percentiles as well as the median (horizontal line) and the mean (+). pg/mL: picograms per milliliter.

**Table 1 animals-13-02676-t001:** Descriptive statistics of interleukin-31 (IL-31) levels in serum and cerebrospinal fluid (CSF) samples. The evaluation of the 95% confidence interval of mean was not possible for groups C and D due to an insufficient sample size.

Group		Type	Number	Mean IL-31 Content (pg/mL)	Minimum (pg/mL)	Maximum (pg/mL)	Lower 95% CI of Mean (pg/mL)	Upper 95% CI of Mean (pg/mL)
A	SRMA	Serum	18	260.7	10.78	1050	103.4	418.0
		CSF	23	183.5	35.95	747.1	113.9	235.1
B	MUO	Serum	15	209.9	0	815.6	92.94	326.9
		CSF	19	218.0	25.07	800.7	109.2	326.9
C	Infectious Meningoencephalitis	Serum	3	98.37	58.13	150.1	Too small	Too small
		CSF	2	240.2	154.9	325.4	Too small	Too small
D	Atopic dermatitis	Serum	3	228.3	62.97	437.9	Too small	Too small
		CSF	-	-	-	-	-	-
E	Healthy control group	Serum	10	80.73	42.67	137.2	53.64	107.8
		CSF	8	186.2	98.63	309.6	129.1	243.3
Total		Serum	49	196.5	0	1050	128.9	264.1
		CSF	52	198.7	25.07	800.7	150.2	247.3

SRMA: steroid-responsive meningitis-arteritis; MUO: meningoencephalitis of unknown origin, pg/mL: picograms per milliliter; CI: confidence interval.

## Data Availability

The data presented in this study are available in Appendix A.

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
