# Peer review of "Elevated Interleukin-31 Levels in Serum, but Not CSF of Dogs with Steroid-Responsive Meningitis-Arteritis Suggest an Involvement in Its Pathogenesis"

_animals, 2023, doi:10.3390/ani13162676_

Round 1

Reviewer 1 Report

This manuscript provides an analysis of some less investigated features of SRMA, recognition of which, may advance understanding of the pathogenesis of this condition. However, some analysis and various statements in the discussion do not seem to be justified on evidence gathered and reference to previous studies fails, on occasion, to deliver the information that has a bearing on the questions examined. Moreover, rationale for some of the analysis is not stated explicitly in the introduction and overall, the MS could use a thorough job of editing to clarify many confusing statements. Even though it is doubtless that the MS research question provides a foundation for future studies on this issue. Authors are encouraged to address issues as:

Table 1: please provide means with 95% confidence intervals instead.

Line 65 and 66: the rationale behind inclusion of the MUO and infectious ME groups in this analysis should appear somewhere in the MS.

Line 306: [..] “The purpose of this study was to investigate the presence of IL-31 in serum and CSF  in patients with SRMA and their dependence on a possible pre-treatment” – this is not stated in the introduction/aims. Please remove this sentence or make changes accordingly.

Line 311: [..] “In SRMA several acute-phase proteins are elevated indicate an inflammatory reaction [12,34]”. Revise style/construct of the sentence please.

Line 320 [..]IL-31 levels were elevated in serum samples of dogs with suspected Th2 cell mediated inflammatory response including dogs with secondary inflammation due to intervertebral disc herniation or with suspected otitis media and interna [..] this part should be better expanded particularly considering that a reference to unpublished study [16] does not allow meaningful assessment of the statement here reported or comparisons with the results of the MS.

Line 329: [..] high levels of IL-31 were observed in serum of SRMA patients with a peracute or acute disease process of three days or less, whereas SRMA patients with low serum IL-31 levels had predominantly shown clinical signs of disease for at least five days. [..] The information, already presented in the results section, it is not further expanded here. It remains unclear to what extent the systemic and intrathecal cytokine response correlates to duration of clinical signs and if/or CSF or serum IL levels differ for dogs pre-treated relatively based on duration of signs.

Line 216 and Line 332: [..] patients with very high IL-31 levels in serum showed increased body temperature in the clinical examination.[..] this information is not clearly detailed in materials and methods or results (i.e. was a “cut off” for IL 31 levels considered? Why IL 31 levels were considered highly elevated only in 3 of 4 of SRMA dogs? What about the other dogs affected by SRMA? Was pyrexia encountered in dogs with infectious meningoencephalitis?)

Line 364: [..] Therefore our results are in line with the previous literature as higher IL-31 levels were found in the Th2-driven immune response in SRMA.[..] as the number of NE, NME in the group denominated “MUO” versus GME is unknown, identifying the immune pathway (Th1 or Th2) involved in current results is impossible, however, dogs in the MUO group appear to have very high IL31 serum levels compared to healthy dogs, and behave similarly to SRMA dogs. The results of this study are not in line with previous literature, no conclusion can be drawn from these results regarding a Th2 or Th1 type of immune response for the MUO group. Please rephrase/change the paragraph to avoid confusing statements.

Line 244 [..] There is a significant difference within the serum samples between the groups B {MUO cases} and E {healthy control group}(p=.0163)[..] and Line 367: [..] Nevertheless, patients with MUO also had elevated IL-31 levels compared to the healthy control group in this study, but lower levels than the dogs with SRMA [..] no statistically significant differences were reported between serum levels of IL31 among dogs with MUO or SRMA, and CSF levels of IL 31 are higher in MUO than in SRMA dogs. It appears that IL31 plays an important role in MUO dogs (and as well in infectious meningoencephalitis cases), please acknowledge and expand this in the discussion and/or clarify what was the initial reason to investigate IL31 levels in the MUO group instead of simply comparing levels in SRMA and healthy dogs.

Line 409[..] After  treatment with prednisolone, the integrity of the blood brain barrier is increased so that the immune cells are trapped within the CSF still producing their interleukins. [..] please provide reference for this statement, eventually clarifying additional points including evidence of survival of immune cells within CSF and timing in IL production after restoration of brain barrier damage during neuroinflammation. Although this may be an interesting theory, it remains speculative and other explanations for such results should be considered. Among these, should be highlighted that current analysis is not supported by good numbers (only 4 dogs have been treated with steroids in this group), due to the retrospective nature of the study (dogs are simultaneously treated with steroids, NSAIDs and other drugs and duration of signs is highly variable) homogenous treatment groups are lacking. Please revise the entire paragraph and clearly acknowledge the limitations of this study in the discussion.

Line 418: please clarify what “systemic changes” are– are these seen/occurring also in MUO?

Line 420:[..] Prospective studies examining the influence on maintenance of SRMA [..] Please remove or rephrase this very confusing and awkward sentence.

The MS could use a thorough job of proof reading and editing to clarify  confusing statements

Reviewer 2 Report

Thank you for the opportunity to review this manuscript. Overall I found your study design to be appropriate and well described, and the discussion thorough. The introduction would benefit from briefly discussing MUO and infectious meningitis and how these differ from SRMA but why they are of value as comparative diseases in the present study; additionally, I ask for clarification to the hypothesis statement. The results section seems to be missing reporting of some of the data. I have provided some comments below of specific areas where I suggest editing or adding clarification. 

Consider editing title to: Elevated interleukin-31 levels in serum but not CSF of dogs with steroid-responsive meningitis-arteritis suggests an involvement in its pathogenesis

Line 16-21: Please add p-values throughout this portion of the abstract. 

Line 57-58: I think additional clarity to the hypothesis statement would be helpful here. Specifically I would like to know how you expected the IL-31 levels to compare to MUO, infectious meningoencephalitis, and atopic dermatitis and what you expected to find in regards to pre-treatment status. 

Line 185-188: What about IL-31 levels in SRMA compared to healthy controls or the other diseases? I’m a little confused why you are only reporting the statistical significance of IL-31 in SRMA compared to atopic dermatitis and not the other diseases or healthy controls. I am also not sure if it is needed to report the significance of IL-31 levels in MUO versus healthy controls as this is not the aim of this study (or edit the hypothesis/aims to allow the reader to understand why you are reporting this). 

Figure 1 helps to answer some of my questions in the comment above but it feels like I am still missing the bottom line here – how did IL-31 serum levels in SRMA compare statistically to each of the Groups B-E (I would like to see all of those p-values reported clearly). 

Can you add a reference to Figure 1A after that first paragraph in section 3.1 (maybe after Line 189)? 

Line 222-232: Again, I’m not sure entirely sure why you include additional information about MUO as this is not the focus of the study. Perhaps this can be clarified with editing the hypothesis/aims. 

Line 237: P-value for CSF IL-31 levels of MUO versus SRMA/healthy controls? 

Line 350: I think you meant “site” not “side of CSF sampling”. Same for Line 351. 

General comment about Discussion: Please discuss limitations of this study (e.g. retrospective, relatively small sample sizes especially when subdividing into pre-treatment groups, not always having a histopathologic diagnosis available, etc.) 

Round 2

Reviewer 1 Report

Many thanks to the Authors for this amended version of the MS and for providing a point‐by‐point response to the criticisms raised. The MS has certainly merit for the relevance of its topic and meets in the current analysis, the suggestions of this Reviewer.